# Primary Care Records of Chronic-Disease Patient Adherence to Treatment

**DOI:** 10.3390/ijerph18073710

**Published:** 2021-04-02

**Authors:** Mireia Massot Mesquida, Josep Anton de la Fuente, Anna María Andrés Lorca, Ingrid Arteaga Pillasagua, Edelmiro Balboa Blanco, Sonia Gracia Vidal, Sara Pablo Reyes, Paula Gómez Iparraguirre, Gemma Seda Gombau, Pere Torán-Monserrat

**Affiliations:** 1Servei d’Atenció Primària Vallès Occidental, Gerència Àmbit Metropolitana Nord, Institut Català de la Salut, Sabadell, 08201 Barcelona, Spain; 2Servei d’Atenció Primària Barcelonès Nord i Maresme, Gerència Àmbit Metropolitana Nord, Institut Català de la Salut, Badalona, 08911 Barcelona, Spain; jadelafuente.bnm.ics@gencat.cat (J.A.d.l.F.); sgracia.bnm.ics@gencat.cat (S.G.V.); 3Centre d’Atenció Primària Santa Coloma 1, Gerència Àmbit Metropolitana Nord, Institut Català de la Salut, Santa Coloma, 08921 Barcelona, Spain; amandres.bnm.ics@gencat.cat; 4Centre d’Atenció Primària Vall del Tenes, Gerència Àmbit Metropolitana Nord, Institut Català de la Salut, Lliçà d’Amunt, 08186 Barcelona, Spain; iarteaga@gencat.cat; 5Centre d’Atenció Primària Sant Quirze del Vallès, Gerència Àmbit Metropolitana Nord, Institut Català de la Salut, Sabadell, 08192 Barcelona, Spain; ebalboa.mn.ics@gencat.cat; 6Direcció d’Atenció Primària Metropolitana Nord, Gerència Àmbit Metropolitana Nord, Institut Català de la Salut, Sabadell, 08201 Barcelona, Spain; spablo.mn.ics@gencat.cat; 7Centre d’Atenció Primària Sabadell La Serra, Gerència Àmbit Metropolitana Nord, Institut Català de la Salut, Sabadell, 08202 Barcelona, Spain; pgomezipa.mn.ics@gencat.cat; 8Unitat de Suport a la Recerca Metropolitana Nord, Institut Universitari d’Investigació en Atenció Primària Jordi Gol (IDIAP Jordi Gol), 08303 Mataró, Spain; gemma.seda@gmail.com (G.S.G.); ptoran.bnm.ics@gencat.cat (P.T.-M.); 9Department of Medicine, Faculty of Medicine, Universitat de Girona, 17004 Girona, Spain

**Keywords:** medication adherence, chronic diseases, primary care, healthcare evaluation mechanisms

## Abstract

The goal of managing adherence (AD) is to achieve better medication use by patients in order to maximize benefits and reduce risks. With the aim of improving treatment adherence by patients, we carried out a descriptive study to obtain information related to adherence management in primary care. Inclusion criteria were as follows: patients that had at least one record of any treatment adherence assessment variable. For those that had more than one recorded variable, we analyzed consistency across test results. For the comparative analysis of adherence records, patients were categorized into three groups on the basis of the healthcare unit that recorded the data: case management (CM), home care (HC), and primary care team (PCT). A total of 32,137 subjects met inclusion criteria; 79.56% of subjects were older than 65. As for the analysis of assessment records across care units, 69.73% of CM patients, 67.17% of HC patients, and 2.33% of PCT patients had adherence assessment records. CM units made a significantly greater number of records than HC units. We observed low adherence at a rate of 49.3% in the CM group, 31.91% in the HC group, and 17.58% in the PCT group. When more than one adherence variable was recorded, analysis revealed inconsistent test results or recorded variables in 9.06% of PCT cases, 14.83% of HC cases, and 20.47% of CM cases. The inconsistencies observed in records of adherence assessment and management across different care units reveal the huge variability that exists in managing and selecting a tool to assess adherence.

## 1. Introduction

Adherence to pharmacological treatment constitutes a primary concern in regular clinical follow-up of patients with chronic diseases given the impact it has on disease control and increased morbimortality. Various studies [1,2,3,4,5], as well as the World Health Organization [6] (WHO), have concluded that medication nonadherence leads to suboptimal clinical results, increased morbidity and mortality rates, and increased healthcare costs. Between approximately 50% and 60% of patients demonstrate poor medication adherence, especially those with chronic diseases. Consequently, 30% of hospital admissions are related to poor adherence [7,8]. By way of example, the results of a meta-analysis [5] of 21 studies (46,847 participants) showed that good adherence was associated with lower mortality as compared to poor adherence (odds ratio 0.56, 95% confidence interval 0.50 to 0.63). Similarly, a study conducted by Ho et al. on patients with diabetes showed that medication nonadherence was associated with higher rates of all-cause hospitalization and all-cause mortality (23.2% vs. 19.2%, *p* = 0.001 and 5.9% vs. 4.0%, *p* = 0.001, respectively) [4].

In response to this situation, in 2003, the WHO [6] published the following definition of adherence according to characterizations by Haynes and Rand [9]: “the extent to which a person’s behavior—taking medication, following a diet, and/or executing lifestyle changes—corresponds with agreed recommendations from a healthcare provider.” Along these lines, in 2011 Vrijens [10] published a new taxonomy for describing and defining medication adherence, in which adherence was defined as “the process by which patients take their medication as prescribed, further divided into three quantifiable phases: ‘initiation’, ‘implementation’ and ‘discontinuation’”. This definition implies cooperation between prescriber and patient. The term “adherence”, thus, differs from “compliance” in that the latter entails subordination of the patient to the prescribing doctor’s instructions. The concept of adherence, however, implies a change in perspective, to one in which the patient’s view is taken into account during decision-making. One must bear in mind that the patient’s perspective is crucial to understanding their attitude toward taking medication and, more specifically, toward treatment adherence following consultation [11].

Terminology aside, it is clear that treatment goals and the benefits linked to prescribed drugs can only be reached if patients are persistent in taking medication over the long term [12,13].

Despite the fact that good treatment adherence is very often crucial to achieving the desired goals or cure for an illness, as of now there is no gold standard for determining nonadherence. Nevertheless, there are multiple tools [14,15] available to determine treatment adherence. These tools come with advantages and disadvantages, mostly due to patient subjectivity when responding to questions. Thus, the general consensus is to use more than one tool to detect nonadherence to pharmacological treatment [16,17,18].

Adherence studies face two significant limitations: firstly, incomplete theory that adequately predicts and explains nonadherence; secondly, inconsistency in the definitions of variables, as well as in the interventions carried out across studies, which, in turn, complicates cross-comparison. In light of these limitations, Helmy et al. [19] initiated a Delphi study to create a guide that promotes quality in clinical research on adherence by reaching a consensus on different variables. By defining these variables, they also aimed to limit the ambiguity that surrounds definitions or criteria used to establish treatment nonadherence.

Despite limitations to the comparison of effective interventions used to improve adherence, the bibliography [20,21,22] presents predicting factors for nonadherence, which the WHO categorizes into five types: socioeconomic factors, factors associated with the healthcare system, therapy-related factors, disease-related factors, and patient-related factors.

The goal of managing adherence is to achieve better medication use by patients in order to maximize benefits and reduce risks, as demonstrated by studies in which action was taken to improve treatment adherence and resulted in a proven positive association between prescribed treatment and health results [23,24].

With the aim of improving treatment adherence by patients with complex chronic diseases within our sphere of influence, we carried out a descriptive study to obtain information related to detection, recording, prevalence, and causes of failure to adhere, according to the healthcare model used for each patient in our primary care practices.

## 2. Materials and Methods

### 2.1. Study Cohort

We carried out a retrospective population-based, cross-sectional, descriptive study on patients assigned to 64 primary care practices within the Barcelona Metropolitan North region (MN) (Catalonia), which provides coverage to 1,349,788 inhabitants.

For the comparative analysis of adherence records, patients were categorized into three groups on the basis of healthcare unit: case management (CM), home care (HC), and primary care team (PCT).

Patients were included in the HC program when, due to physical health conditions or social circumstances, they required social and/or healthcare either temporarily or permanently but were unable to go to the healthcare practices. Patients were included in the CM category when they met clinical complexity profile requirements [25,26,27] as defined by the concurrence of several chronic diseases, multiple emergency or unscheduled hospital admissions and/or visits to emergency room, polymedication, and difficulty performing some daily activities. Similarly, patients included in the PCT program were seen by their primary care physician and/or nurse and were not included in any other category.

It was possible for a given patient to be included in two of these categories since they were not exclusive. Nevertheless, for the study, patients were included only in the most complex category to which they could be attributed.

The study population extracted from electronic clinical records (ECR) consisted of patients assigned to participating primary care practices who had at least one record of assessment of adherence to pharmacological treatment during a 1 year period. Patients who did not meet inclusion criteria were excluded from the study.

We used the ECR from primary care, which include the demographical, clinical, and prescription-drug data of patients assigned to the 64 primary care practices that participated in the study. The information contained in the ECR is recorded by primary care doctors and nurses in their clinical practice. In Catalonia, primary care medical records have been digitalized for over 15 years and can be accessed at all primary care practices.

### 2.2. Adherence Measures

There are a variety of quantitative and qualitative tools available to assess adherence. The choice of which tool or combination of tools to use is left to the clinical judgment of healthcare professionals. Therefore, the variables used to identify adherence (AD) assessment records included any validated test for measuring medication adherence [16,17,18] and/or qualitative variables (QV) that classified adherence as adequate, moderate, or low on the basis of the doctor’s clinical criteria. Table 1 describes different clinical criteria and the corresponding value that general practitioners (GP) might use to determine whether or not a patient is adhering to their prescribed treatment.

Moreover, a computerized data sheet was available to record the cause of the patient’s failure to adhere. For those that had more than one recorded adherence variable during the study period, we analyzed consistency across test results. Adherence was considered to be low when QV results were moderate or low. In the case of validated tests for measuring medication adherence, the cutoff criteria of the tests determined the degree of adherence.

### 2.3. Covariables

The following covariables were taken into account for the analyses: age, gender, and whether or not subjects were considered to be chronic complex patients (CCP), to have advanced chronic disease (ACD), or to have chronic disease according to the adjusted morbidity group (AMG) [28]. The AMG classifies users on the basis of disease type (severe, chronic, or oncological) and, in the case of chronic disease, whether or not the user presents multimorbidity.

Chronic complex patients (CCP) are those who require special care plans due to a clinical situation that is difficult to manage, often as a result of the accumulation of concomitant chronic diseases accompanied by an intense use of resources, especially hospitalization, which could often be avoided. Patients with advanced chronic disease (ACD) are those that have one or more chronic diseases in an advanced, severe, or progressive stage, a strong need for care, and often limited life expectancy.

### 2.4. Statistical Analysis

We carried out a descriptive analysis of the variables studied, as well as of the number and combination of variables used in adherence assessments. When more than one variable was used, we studied the degree of consistency across results from tests/recorded variables. If two or more adherence assessment tests or measurements produced the same results, we considered them to be consistent. We compared results across three healthcare units (CM, HC, PCT) using Pearson’s chi-squared test for qualitative variables and the Student’s *t*-test for quantitative variables.

## 3. Results

Prevalence of chronic disease (bearing in mind the AMG) within the entire population of 1,349,788 subjects in the MN region was 77.3% (range 77.4–80.2%). A total of 32,137 subjects met inclusion criteria and were included in the study (Figure 1), which represents 2.38% of the total population and 3.08% of patients with chronic diseases. Mean age was 75.7 (SD: 12.6); 79.56% of subjects were older than 65 and 50.4% were women. Table 2 presents the distribution of patients included in the study according to typology (CCP and ACD). Adherence assessment percentages were greater in CCP and ACD patients than in noncomplex chronic patients (Table 3).

The global percentage of low adherence to pharmacological treatment, taking into account the results of validated tests and qualitative variables (QV), is 17.88 in the studied population. In the analysis of low adherence across care groups, the differences observed were statistically significant (*p* < 0.001) (Table 3).

Regarding the analysis of patient typology across care units, adherence assessment for CCPs was 77.73% in CM, 67.17% in HC, and 42.10% in regular clinical PCT practice (*p* < 0.001) (Table 4).

The variables most often used to assess AD across all care units were QVs, which were employed in 76.70% of cases, while validated tests were only used 22.66% of the time as the sole recorded variable. In the analysis by care units, of the 24,722 patients that were assessed for adherence to pharmacological treatment by PCTs, only one adherence variable was recorded for 85.12% of patients, two variables were recorded for 11.20% of patients, and three variables were recorded for 3.40% of patients. When more than one variable was used, the most common combination (in 7.86% of cases) was the use of two validated tests together with a variable for global assessment of adherence to pharmacological treatment. In only 3.46% of cases, test results were accompanied by a recorded QV representing the cause of the patient’s failure to adhere to pharmacological treatment. The HC unit carried out assessment most often (in 54.79% of patients) with just one variable (even though this rate was lower than that observed in PCT units), while assessment using two variables was observed in 33.43% of cases. In CM, adherence assessment was most often carried out by recording two variables (50.51%). When just one variable was used, the HC unit followed the same pattern as PCT units using a QV for adherence assessment, while the CM unit relied on validated tests more often (52.78%).

When more than one adherence variable was recorded, analysis revealed inconsistency of test results or recorded variables in 9.06% of PCT cases, 14.83% of HC cases, and 20.47% of CM cases.

## 4. Discussion

The results of this study reveal significant variability across care units in the recording of adherence assessments. The PCT care unit presented the lowest percentage of adherence records. Differences in records may be due to the typology of patients treated by these teams. According to the bibliography [3,21,29,30], the factors that affect adherence are complex and disputable, since various factors can interact in a single patient and, thus, cause undetermined effects. However, age, polymedication, chronic or mental illness, memory problems, and functional capacity issues, among others, have been identified as determining factors or predictors of low adherence to pharmacological treatment [6,22]. Other important aspects to consider include the treatment burden and the complexity of pharmacotherapy. Treatment burden refers to the workload imposed by healthcare on patients and the effect this has on quality of life [31,32]. The intensity of treatment burden varies for each patient depending on their situation, but those with complex social situations are the most vulnerable. As for the complexity of pharmacotherapy, a higher number of medications and complicated schedules or special instructions (e.g., time of day, food interactions) can all contribute to greater patient difficulty or lower interest in following treatment recommendations. Greater complexity is linked to lower treatment adherence [33]. Prevalence of these determining factors in patients included in HC and CM programs is very high in comparison to patients attended directly by PCTs. This may foster the predisposition of HC and CM professionals to assess adherence, given that they treat patients at higher risk for low adherence to pharmacological treatment. The focus of these units on chronicity may also be a predisposing factor, as they seek out effective management of medication with the aim of preventing hospitalization or emergency visits related to failure to adhere to prescribed medication [3,12]. In a systematic review, Al Hamid et al. [8] concluded that the adverse effects of medication and failure to adhere to medication were the main causes of hospitalization. In the case of PCT units, the workload burden may diminish the perceived importance of assessing adherence to pharmacological treatment.

Regarding variability in the prevalence of low adherence to pharmacological treatment across the care units evaluated, it was highest in the CM unit (almost three times higher than in the PCT unit). This may be due to the fact that patients included in CM are at higher risk for low adherence due to greater prevalence of determining factors or predictors for low adherence in this group. Another possible explanation for this difference is the assessment methodology used by CM nurses to assess treatment adherence. As mentioned previously, there are a variety of quantitative and qualitative tools available to assess adherence, leaving the choice of which tool or combination of tools to use up to healthcare professionals. According to various sources in the bibliography [14,15,16], the general consensus is that the choice of tool should be based on scientific evidence used to create it, what it measures, and how it has been validated, since the majority of tools have been validated for specific illnesses and not for the assessment of patients with multiple illnesses [34]. Use of more than one tool is also recommended to determine adherence due to the lack of a gold standard and patient propensity to overestimate adherence when asked about it. In the CM unit, adherence assessment, as reflected by the results, was most often carried out using a combination of two strategies: a validated test and review of the patient’s medicine cabinet, since CM and HC nurses go to patients’ homes to assess and treat them. This method is recommended by different authors [15,16,17,18,34,35,36] who endorse combining a self-administered test with a more pragmatic yet indirect method, such as counting pills, medication dispensed at the pharmacy, or biomedical parameters used to determine the degree of control over the illness. Nevertheless, PCT units, as indicated by the results, most often used just one tool to assess adherence. The use of a sole recorded variable becomes a limitation to providing clear and objective assessment, given that, in this unit, QVs were used more often than a test, which are validated and/or based on science, unlike QVs. This qualitative variable depends on the clinical criteria of the observer and, thus, carries a great deal of variability in results. Greater use of QVs may be due to the fact that they are easy and quick to use in high-pressure healthcare settings. Infrequent recording of causes and determining factors for low adherence to pharmacological treatment (which the WHO [6] categorizes into five groups: socioeconomic factors, factors associated with the health care system, therapy-related factors, disease-related factors, and patient-related factors) is due to the fact that this record is not available in the entire region, thus making its use sometimes impossible.

The prevalence of low adherence to treatment observed across units is not comparable to the results of other studies, since the majority refers to the prevalence of low adherence in the case of specific diseases and not to polymedicated patients with multiple chronic diseases. Nevertheless, in a study that assessed global adherence to pharmacological treatment and in which the characteristics of patients included were similar to those in HC or CM units, Turner et al. [29] found that 90% of patients presented low adherence to at least one medication of their pharmacotherapy and, on average, only 58% of patients adhered to prescribed treatment.

Regarding the contradiction in results when more than one tool or test was used to assess adherence, Marcum et al. [34] also found a high degree of contradiction across adherence assessment results. Only 3.7% of patients presented consistent results. Inconsistency across results is due to the use of two tests that employ very different strategies to measure adherence to pharmacological treatment. Validated tests use the technique of directly asking the patient, while QVs allow medical professionals to review patients’ medicine cabinets, analytical variables, or disease control variables. As mentioned before, this underscores the necessity of using more than one tool, which, when possible, should be complementary or measure different dimensions to assess adherence in polymedicated patients with multiple chronic diseases.

One of the strengths of this study is that it is a descriptive analysis of records of different variables in electronic medical records and, thus, provides a deeper look at the adherence activity carried out in primary care practices. It was conducted on an extensive primary care population and constitutes a comprehensive review of treatment adherence management by primary care doctors and/or nurses.

As for its limitations, because this study examines different adherence assessment measurements used by healthcare professionals (different tests and qualitative variables), it may lead to variability in the clinical assessment of adherence. This can be observed when analyzing the consistency of different tests used on a single patient. Moreover, since this study uses a database of clinical records, there are no individual data available on the healthcare professionals that can be used to analyze the impact of these factors on their examination of adherence or the tool they use. Lastly, low activity records or records that do not faithfully represent this activity may constitute a limitation of this study.

Despite these limitations, we believe that the approach taken in this study to analyze adherence management in primary care provides valid information about regular clinical practice.

## 5. Conclusions

The inconsistencies observed in records of adherence assessment and management across different care units reveals the huge variability that exists in managing and selecting a tool to assess adherence between different primary care units. Given this variability, we will develop an improvement plan in our primary care practices to unify the definition of adherence and the tool(s) used to assess it, as well as to modify clinical workstations so that assessment action can be properly recorded. Awareness intervention is also needed to increase the examination of adherence in primary care units and to increase actions to improve adherence in patients with complex chronic disease. Tools to assess patient centered adherence with multiple illnesses need to be developed. The aim of this study was to understand how primary care professionals record adherence in ECR. Further research should be conducted to explore differences on the basis of the characteristics of the patient (age, sex, comorbidities, etc.) and professionals, as well as the tool used.

## Figures and Tables

**Figure 1 ijerph-18-03710-f001:**
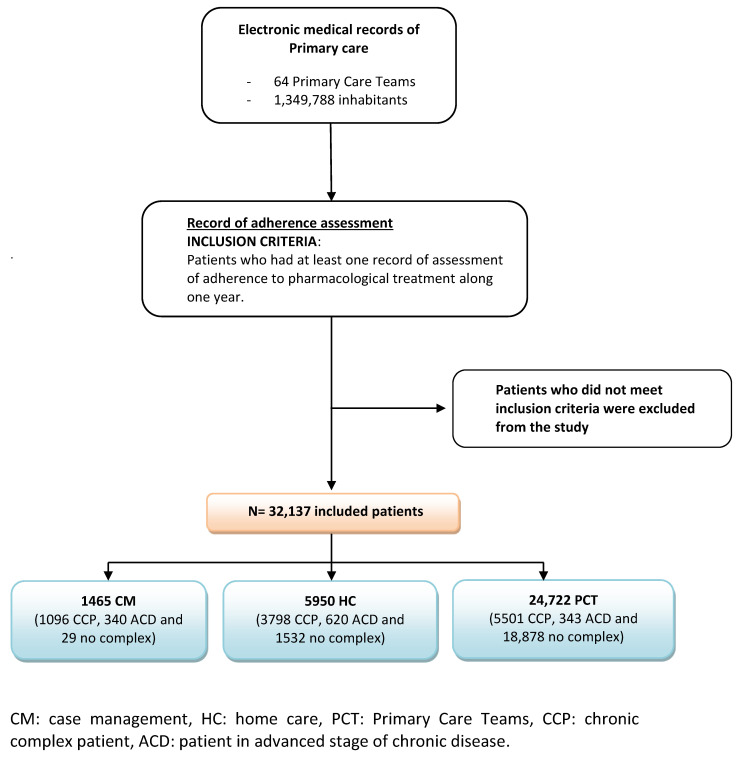
Flow chart of patient inclusion at each point of the study. Abbreviations: CM: case management, HC: home care, PCT: primary care teams, CCP: chronic complex patient, ACD: patient in advanced stage of chronic disease.

**Table 1 ijerph-18-03710-t001:** Clinical criteria and corresponding value indicating failure to adhere.

Clinical Criteria	Value Indicating Failure to Adhere
Medication possession ratio *	Less than 80%
Effectiveness of additional medications	Ineffective after adding 1–2 medications to current treatment
Clinical interview	Patient confirmation or explanation
Medicine cabinet review	High number of full boxes

* Medication possession ratio calculation is made during the appointment by the doctor or nurse. Both the doctor and the nurse have access to, on the one hand, the medication prescribed, and, on the other hand, the registry of medication removed from pharmacies (number of packages and date of collection).

**Table 2 ijerph-18-03710-t002:** Distribution of recorded adherence assessments based on patient typology.

Patient Typology	Total Patients (*n*)	Patients Included (*n*)	Adherence Assessment (%)
CCP	20,131	10,395	51.54%
ACD	3986	1303	32.69%
Noncomplex	1323,670	20,439	1.54%

CCP: chronic complex patient, ACD: patient in advanced stage of chronic disease.

**Table 3 ijerph-18-03710-t003:** Distribution of adherence records and percentage of failure to adhere based on care unit.

Care Unit	Total (*n*, Chronic Patients)	*n* Patients ^(1)^	% Adherence Assessment	% Failure to Adhere
HC	8858	5950	67.17% ^+^	31.91% *
CCP not CM		3798		
ACD not CM		620		
Noncomplex		1532		
CM	2101	1465	69.73% ^+^	49.3% *
CCP		1096		
ACD		340		
Noncomplex		29		
PCT	1,043,386	24,722	2.37% ^+^	17.58% *
CCP not CM		5501		
ACD not CM		343		
Noncomplex		18,878		

HC: home care, CCP: chronic complex patient, CM case management, ACD: patient in advanced stage of chronic disease, PCT: primary care teams. ^+^ CM units made a significantly greater number of records than HC units (*p* = 0.03) In regular clinical PCT practice, the percentage was significantly lower in comparison to CM and HC (*p* < 0.001). ^(1)^ Patients who had at least one record of assessment of adherence to pharmacological treatment along one year. * In the analysis of low adherence across care groups, the differences observed were statistically significant (*p* < 0.001).

**Table 4 ijerph-18-03710-t004:** Distribution of adherence assessment in CCPs according to care unit.

	Total *n* of CCPs	No. of Patients with AD Assessment	%
CM	1410	1096	77.73%
HC	5654	3798	67.17%
PCT	13,067	5501	42.10%

## Data Availability

The datasets generated and/or analyzed during the current study are not publicly available due politics of our institution but are available from the corresponding author on reasonable request.

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
