# Peer review of "Primary Care Records of Chronic-Disease Patient Adherence to Treatment"

_ijerph, 2021, doi:10.3390/ijerph18073710_

Round 1

Reviewer 1 Report

The paper by Dr. Massot Mesquida aims to (i) assess the consistency across test results among patients who had more than one recorded adherence assessment, and (ii) compare the proportion of non-adherent patients across health care units. They observed inconsistent test results ranged from 9% (PCT patients) to 20% (CM patients), and a huge difference in low adherence rate across groups (49% among CM patients and 18% among PCT patients). I believe this manuscript is worthy because it adds significant evidence concerning the global adherence to pharmacological treatment (compared to studies that evaluate adherence to specific drug therapies).

The paper raises some comments.

  • I suggest organising the Methods section in paragraphs with the following headings: Study cohort (including the categorization into the three groups), Adherence measures, Covariates, Statistical Analysis.
  • I wonder about the representativeness of the study sample. The 1,349,788 inhabitants covered by the 64 Primary Care Practices are representative of the entire Barcelona Metropolitan North region population? Again, are patients who had at least one record of assessment of adherence representative of the entire population?
  • Because the inconsistency between test results is one of the main findings of this manuscript, I kindly ask the Authors (i) a clearer definition of “consistency”, and (ii) a deeper evaluation of this phenomenon. Regarding the latter request, I suggest calculating the Cohen's kappa coefficient and evaluate some predictors of inconsistent results (i.e., this proportion varies across adherence measures, age/sex strata, etc.?).
  • Authors should add some statistics of age and sex distribution among groups.
  • Why Authors did not perform a statistical model to assess differences in adherence between groups taking into account the tool used to assess adherence, patient’s (age, sex, chronic complexity) and physician’s characteristics?

Author Response

Dear Reviewer 1, 

Reviewer 2 Report

Overall comments

This manuscript conducts a retrospective population-based, cross-sectional, and descriptive study in order to gain knowledge about detection, recording, prevalence and the causes of adherence failure. It is recommended to significantly revise the manuscript based on the comments below in order to improve the overall research and readability of this manuscript.

Specific comments

1. Introduction

  • Please add any previous findings on why adherence in primary care is important.
  • Line 65. The difference between “adherence” and “compliance” is described here. Further explanation is needed: why does the difference need to be declared? What kind of outcome could be derived from “imply the patient’s view”?
  • Line 91. Please use a consistent paragraph form.

2. Materials and Methods

  • Overall, please rewrite the Materials and Methods section so that it is more scientific. This section was unclearly written.
  • Line 128. The meaning of the first sentence is not clear. If it is necessary to mention the reason for taking home care (“due to physical health conditions or social circumstances”), please explain that idea in a separate sentence.
  • Line 131. Please add references about “clinical complexity profile requirements”.
  • Line 146. Please add an explanation of Adjusted Morbidity Group.

3. Results

  • Overall, the Results section should be rewritten to clarify the study results.
  • Line 154: please explain what diseases were classified as chronic diseases.
  • Please explain how percentage of adherence was calculated and what method of calculating adherence was used.
  • Line 164. In Figure 1, there are a total of 1,495 individuals in CM, 5,950 individuals in HC, and 20,439 individuals in PCT. But the sum of these totals is 27,884 not 32,137. Please resolve this discrepancy. In addition, except for the HC group, the sub-sequential number of both CM and PCT groupS are not matched. For example, 1,096 CCP+340 ACD=1,436≠1495; 5,501 CCP+343 ACD+18,878 no complex=24,722≠20,439. Please specifically explain these discrepancies in the manuscript.
  • Line 167. The numbers in Table 2 do not match the numbers in Figure 1. According to Figure 1, there are a total of 20,410 “non-complex” patients included in the study. The number in Table 2 is 20,439.
  • Line 171. Please confirm the following sentence is correct: “+CM unites made a significantly greater number of records than HC units”.
  • Line 177. Please explain how the global percentage of low adherence (23.80%) was calculated.

4. Discussion

  • The discussion section should be scientifically rewritten including study limitations.
  • Line 224. The authors provide two explanations for the reason of low adherence to pharmacological treatment in the CM unit. One is that patients in the CM group are facing higher risk for low adherence. Another is the assessment methodology used by CM nurses. Both statements need to be enriched either by citing other references or by including additional explanations as to their respective contributions to low adherence.

Author Response

Dear Reviewer 2, 

Round 2

Reviewer 2 Report

Authors should describe how they calculated medication possession ratio and assessed adherence. 
